# Deregulated *miR-29b-3p* Correlates with Tissue-Specific Activation of Intrinsic Apoptosis in An Animal Model of Amyotrophic Lateral Sclerosis

**DOI:** 10.3390/cells8091077

**Published:** 2019-09-12

**Authors:** Christina L. Klatt, Verena Theis, Stephan Hahn, Carsten Theiss, Veronika Matschke

**Affiliations:** 1Ruhr University Bochum, Medical Faculty, Institute of Anatomy, Department of Cytology, 44801 Bochum, Germany; christinaklatt@online.de (C.L.K.); verena.theis@rub.de (V.T.); carsten.theiss@rub.de (C.T.); 2Ruhr University Bochum, Clinical Research Center, Department of Molecular Gastrointestinal Oncology, 44801 Bochum, Germany; stephan.hahn@rub.de

**Keywords:** ALS, wobbler mouse, apoptosis, microRNA, neurodegeneration, cerebellum, spinal cord, motor neuron disease

## Abstract

Amyotrophic lateral sclerosis (ALS) is one of the most common incurable motor neuron disorders in adults. The majority of all ALS cases occur sporadically (sALS). Symptoms of ALS are caused by a progressive degeneration of motor neurons located in the motor cortex and spinal cord. The question arises why motor neurons selectively degenerate in ALS, while other cells and systems appear to be spared the disease. Members of the intrinsic apoptotic pathway are frequent targets of altered microRNA expression. Therefore, microRNAs and their effects on cell survival are subject of controversial debates. In this study, we investigated the expression of numerous members of the intrinsic apoptotic cascade by qPCR, western blot, and immunostaining in two different regions of the CNS of wobbler mice. Further we addressed the expression of *miR-29b-3p* targeting BMF, Bax, and, Bak, members of the apoptotic pathway. We show a tissue-specific differential expression of BMF, Bax, and cleaved-Caspase 3 in wobbler mice. An opposing regulation of *miR-29b-3p* expression in the cerebellum and cervical spinal cord of wobbler mice suggests different mechanisms regulating the intrinsic apoptotic pathway. Based on our findings, it could be speculated that *miR-29b-3p* might regulate antiapoptotic survival mechanisms in CNS areas that are not affected by neurodegeneration in the wobbler mouse ALS model.

## 1. Introduction

Amyotrophic lateral sclerosis (ALS) represents the most common motor neuron disease in adults [1]. ALS depicts a fatal neurodegenerative disease that inevitably leads to death within two to five years after diagnosis. The clinical phenotype is based on an irreversible and progressive degeneration of motor neurons located in the motor cortex as well as in the anterior horn of the spinal cord [2]. Degeneration of the upper motor neurons manifests as spasticity, whereas a decline of the lower motor neurons is accompanied by flaccid paresis [3]. Therapy possibilities are currently limited to Riluzole and Edavarone. However, both drugs only slightly improve the lifespan by a few months [4,5].

Despite intensive research, the cause of the disease remains still unclear. A familial, genetic background is only detectable in about 10% of ALS patients (fALS), whereas the majority of the cases occurs sporadically, without familial accumulation (sALS) [6]. Aggregations of misfolded proteins, axonal transport disorders, mitochondrial dysfunction, glutamate-induced excitotoxicity, neuroinflammation, and altered RNA processing represent pathological hallmarks of ALS [7].

In this study, we used the wobbler mouse (WR) as an animal model for ALS. The wobbler mouse mutation arose spontaneously in a C57BL/Fa mouse strain and was first characterized by Falconer in 1956 [8]. A single base pair exchange from leucine to glutamine in the last exon of vacuolar-vesicular protein sorting factor 54 (Vps54) was identified to result in a loss-of-function mutation [9]. As a consequence, Vps54 lacks stability, which engenders destabilization of the whole Golgi-associated retrograde protein (GARP) complex, that physiologically regulates retrograde transport from endosomes to the trans-Golgi network [10]. Located in mouse chromosome 11, the affected region corresponds to 2q13-14 in the human genome [11]. The overwhelming advantage of the wobbler model is the display of almost all clinical hallmarks of human ALS patients [12,13]. The sole symptom the mice are lacking is frontotemporal dementia, which appears in roughly 10% of human ALS cases, and could not be evaluated in the mouse model [14]. Furthermore, male mice display a defect of spermatogenesis that mirrors in their infertility, whereas human patients are not affected by impaired reproduction [15]. However, the striking resemblances appear not only on macroscopic, but also on the cell level. Shared pathologies are impaired axonal transport, ubiquitin-positive protein aggregates, degeneration of upper and lower motoneuron, and enlarged endosomes vacuolization, just to name a few [12,16,17,18,19]. Earlier criticism of the mouse model targeted the assumption that no mutations in the Vps54 gene were found in human ALS genomes, an argument that has recently been disproved. Alterations with moderate or even high impact in protein expression were characterized within the sequencing project MinE (http://databrowser.projectmine.com/). Admittedly, the uninterrupted chain of evidence between GARP malfunction and motor neuronal degeneration is still missing.

The wobbler mouse shows ALS-typical symptoms which include damage to the motor system [20]. Even though the cerebellum does not initiate any movements, its influence on fine-tuning, precision, and timing of motion sequences is crucial. Although ALS is considered as a multifactorial disease [21,22,23], the cerebellum shows no signs of neurodegeneration in the three-layered cortex besides some marginal alterations like p62-positive cytosolic inclusions as well as minimal astro- and microgliosis [24,25,26]. However, structural and functional changes of the cerebellum have been demonstrated [27,28,29,30]. This can be explained by the recruitment of the healthy extrapyramidal motor system as a balancing mechanism for the increasing deterioration of motor neuron function [28,31,32]. Notably, the question arises why predominantly motoneurons in the motor cortex as well as in the ventral horn of the spinal cord seem to fail in coping with the wobbler mutation of ubiquitously expressed Vps54, while other neurons, like granule cells and Purkinje cells of the cerebellum, being part of the motor control and motor learning system, seem not to be affected.

The progression of the disease in the wobbler mice is generally divided into three successive stages: the presymptomatic phase (p0), the evolutionary phase (p20), and the stable phase (p40) [33,34]. During the first three postnatal weeks (from p0 to p20), homozygous wobbler mice do not show any clinical abnormalities and no differences between WR and wildtype (WT) littermates can be observed [34]. After p20, the first symptoms, such as head tremor, unsteady gait or muscle weakness of the front limps start to arise [12,33]. At p40, the WR phenotype is fully developed and stagnate [25]. In this study, we chose to solely focus on the presymptomatic and the stable phase. These stages can be seen as starting point and clinical endpoint, the two consistent extremes during the disease’s progress.

Previous work has shown clear evidence that a programmed mechanism of cell death similar to apoptosis is responsible for motor neuron degeneration in ALS [20,35,36,37]. In this study, we suspect that the process of programmed cell death involving deregulated miRNAs may be responsible for the development of the wobbler phenotype, which exhibits much of the clinical symptoms in human ALS patients [12,13,20].

MicroRNAs (miRNAs) are a group of short, highly conserved, noncoding RNAs, which play a crucial role in the post-transcriptional gene regulation of eukaryotes [38]. About 60% of the human genome is influenced by highly specific binding to complementary sequences in the 3′-UTRs of target mRNAs, resulting in their repression or decay [39,40]. Dysregulation of miRNA-related pathways in the central nervous system can lead to severe neuronal impairment and even cell death [41,42,43]. Thus, altered miRNA levels were observed in a variety of neurodegenerative diseases such as Alzheimer’s and Parkinson’s disease, but also ALS [44,45,46,47,48]. Since physiological miRNA expression plays an important role in the maintenance of cell homeostasis and neuronal survival, the importance of these molecules is unavoidable [49,50]. It even has been recently shown that dysregulations of ALS related miRNA levels even continue to manifest in motor neuron progenitor cells derived from induced pluripotent stem cells of fALS and sALS [51]. Consistently, the majority of altered genes in ALS, such as TDP-43, FUS, SOD1, or C9ORF72, are involved in RNA metabolism and processing [45,52]. This may be related to the global downregulation of microRNAs as a common thread of different ALS forms [53,54]. Interactions of TDP-43 and FUS with Drosha and Dicer naturally increase miRNA production through an increased formation of mature miRNA from the precursor miRNA [55,56]. Expression of TDP-43, FUS, or SOD1 mutant proteins attenuates Dicer complex activity either directly or through induction of stress, eventually causing impaired miRNA biogenesis with a harmful overall reduction in miRNA levels [53,57]. The substantial downregulation was demonstrated to be restricted to motor neurons [53,54]. Furthermore, through interaction with proteins involved in RNA metabolism, TDP-43 and FUS enhance the assembly of stress granules. These granules interfere with RNA metabolism, contributing to motor neuron death [58,59,60]. The complex interplay of stress response, mutations, and miRNA disbalance is still far from fully understood. According to the findings of Rizzuti et al., the most affected deregulated pathways relate to ALS relevant mechanisms, including synaptic vesicle regulation, degradation, or apoptosis [51]. Several other groups support their findings with an abundance of differently expressed miRNAs involved in the regulation of neuronal cell death [61,62]. For example, miR-375 seems to protect neurons from DNA-induced stress and support neurogenesis [63]. Intriguingly, low levels of miR-375 could be detected in motor neurons of SMA and in ALS iPSCs correlating with increased p53 expression and other miR375-predicted targets [63,64]. Moreover, a variety of miRNAs were shown to be involved in p53 regulation, a well-characterized proapoptotic protein. Admittedly, the matter is far more complex than failed antiapoptosis. For instance, miR-155 was found to display both pro- and antiapoptotic features, the latter especially via Bcl2 inhibition [65]. Its increased expression was described in the spinal cord of ALS patients and model mice, but also in SOD1 microglia, where they aggravated neuroinflammation [66]. Members of the intrinsic apoptotic pathway, predominantly members of the Bcl-2 family, appear to be frequent targets of altered microRNA expression [67,68]. Under physiological conditions, anti- and proapoptotic players are in an equilibrium [69]. In the presence of pathological stimuli, this carefully balanced homeostasis shifts towards cell death, inducing activation of proteins such as proapoptotic Bax and Bak [70]. This leads to an oligomerization of these proteins and thus to a permeabilization of the mitochondrial outer membrane via pore formation. Therefore, Cytochrome C leaks from the inner membrane of the mitochondria and subsequently activates Caspase 9 that in turn activates Caspase 3 and 7, enabling the proteolytic caspase cascade that ultimately results in cell death [71,72]. It is therefore a meticulous and vulnerable system in which minor changes can easily have devastating consequences for the affected cell [73,74,75].

So far, differential expression of miRNAs and their impact on cell survival through targeting members of the apoptotic pathway is still subject of controversial debates. A member of the *miR-29* family, *miR-29b-3p*, is a strongly disputed candidate. In this study, we choose to focus on the proapoptotic targets of miR-29b-3p: BMF and Bax [72,76,77]. Cellular stress disintegrates a complex consisting of BMF and myosin V, causing the release of the proapoptotic factor. BMF then binds to the antiapoptotic agent Bcl2 and thereby initiates, among other factors, the mechanism of programmed cell death [78]. Otherwise, Bcl2 would interfere with the proapoptotic Bax and prevent an initiation of apoptosis [79]. In addition, BMF causes a dislocation of the proapoptotic factors BIM/PUMA to from a complex with antiapoptotic factors, including Bcl2. BIM/PUMA then have an activating effect on Bak and initiate apoptotic cell death [78]. Even though BMF does not act as a strong initiator of cell death, it depicts a fine-tuning death-inducer [78,80]. The proapoptotic factor Bax is located in the cytosol under physiological conditions but becomes attached to the mitochondrial membrane once triggered by pathological stimuli [81]. As a result, an oligomeric pore in the mitochondrial membrane is formed [82] which causes subsequent Cytochrome C release, initiating the death determinant caspase cascade [83]. As miR-29b-3p disturbs these two proapoptotic factors in its translation, it appears to have an antiapoptotic role. In addition, miR-29b-3p also binds mRNA from antiapoptotic factors, such as Mcl1, providing a more proapoptotic effect. Mcl1, a member of the Bcl2 family, binds proapoptotic factors such as Bak, BIM, or PUMA and thus prevents the initiation of apoptotic cell death [84,85,86]. Since miR29b-3p regulates both, pro- and antiapoptotic proteins (e.g., Bax, Mcl1 or BMF), it is not clear whether deregulation of this miRNA exerts protective or destructive functions [84,87].

Since the Vps54 mutation is present ubiquitously in the wobbler organism, but neurodegeneration mainly occurs in motor neurons, our study aimed to investigate whether a protective mechanism in other CNS regions reduces their tendency to apoptosis. Our findings could prove that different regulatory mechanisms, which restrain apoptosis, take place in the cerebellum of wobbler mice but are absent in the spinal cord.

## 2. Materials and Methods

### 2.1. Animals

All procedures were conducted under established standards of the German federal state of North Rhine Westphalia, in accordance with the European Communities Council Directive 2010/63/EU on the protection of animals used for scientific purposes. Animal experiments were conducted according to the German animal welfare regulations and approved by the local authorities (registration number Az. 84-02.04.2017.A085). Breeding and genotyping of mice was performed as previously described [34]. Wildtype (WT) and wobbler (WR) cervical spinal cord as well as cerebellar tissue was collected at two different ages (presymptomatic phase (p0), and stable symptomatic phase (p40)) and used for further experiments.

### 2.2. RNA Isolation, Reverse Transcription and Quantitative PCR

Isolation of total RNA (tRNA) from cerebellar and spinal cord tissue of healthy wildtype mice and wobbler mice of two different ages was performed using NucleoSpin miRNA Kit (#740971, Macherey-Nagel, Düren, Germany) according to manufacturer’s protocol. cDNA synthesis for mRNA amplification was performed using qScript™ cDNA SuperMix (#95048, Quanta, Beverly, MA, USA), according to the manufacturers protocol using 0.5 µg total RNA and oligo(dT) primer. The reaction mixture was incubated as follows: 5 min at 25 °C, 60 min at 42 °C, stopped by heating at 85 °C for 5 min following an incubation at 4 °C for 5 min. cDNA was stored at −20 °C until use. cDNA synthesis for microRNA amplification was performed using miRCURY LNA™ Universal RT (#203301, Exiqon, Vedbæk, Denmark), according to the manufacturers protocol using 10 µg tRNA. The reaction mixture was incubated for 60 min at 42 °C, inactivated for 5 min at 95 °C, cooled down to 4 °C. The cDNA was stored at −20 °C until use.

Standard quantitative real-time PCR for mRNA analysis was performed on a CFX96 Real-Time PCR Detection System (Bio-Rad, Hercules, CA, USA) using GoTag qPCR Master Mix (#A6001, Promega, Madison, WI, USA). 200 ng of cDNA and 0.7 µM concentrated forward and reverse primer were used. Specific primer for the housekeeping gene *GAPDH* were sense: 5′-CCA GTT CCA TCG GCT TCA TA-3′ and antisense: 5′-CTG TTC AGG GCG AGG TTT-3′. Primer sequences for detection of *BMF* were sense: 5′-AGA ACA CCC AGC CCA TTT AC-3′ and antisense: 5′-GAG GCT TTC TCC CAC CTT TC-3′. Standard quantitative real-time PCR of microRNA was performed using 4 µL of GoTag qPCR Master Mix, 1 µL of undiluted PCR primer mix and 4 µL of cDNA (1:40 dilution) per well. Specific target sequence of hsa-*miR-29b-3p* (#204679, Exiqon) was 5′-UAGCACCAUUUGAAAUCAGUGUU-3′ which is the same as for murine-*miR-29b-3p*. U6 snRNA (has, mmu) miRCURY LNA miRNA (#339306, Exicon, Vedbæk, Denmark) was used as a control for miRNA analysis.

Melting curves were obtained after each PCR run and showed single PCR products. Expression levels for the genes of interest and for the housekeeper *GAPDH* and *U6* were measured in three independent PCR runs in triplicates. The obtained data was analyzed using the 2^−ΔΔCT^ Method [88]. Data analyses were performed with GraphPad Prism 6.1 software (GraphPad Software, La Jolla, CA, USA). Data are provided as means ± SEM. Significant differences were tested using Student’s t-test and results with *p* < 0.05 were considered statistically significant.

### 2.3. In Situ Localization of MicroRNA in Cerebellum and Spinal Cord

In situ hybridization was performed following the instruction manual FFPE in situ hybridization using double-labeled Fluorescein or DIG miCURY LNA™ microRNA Detection probes (Exiqon). Animals at p40 were decapitated and dissection of the central nervous system was performed and stored immediately at −80 °C or instantly sliced. Sixteen micrometer thick slices (Leica AM 3050 S, chamber and stage temperature of −18 and −20 °C, respectively) of wildtype- and wobbler mice cerebella and spinal cord were cut from mice under sterile and RNase-free conditions. Cryosections were fixated with 4% PFA at RT for 30 min, afterwards washed three times for 3 min in PBS and then incubated with 1 µg/mL proteinase-K-buffer-mixture (microRNA ISH Buffer Set, #9000, Exiqon) for 10 min at 37 °C. After removing of the enzyme-mix, the hybridization mix contending 80 nM double-DIG LNA™ microRNA probe (#616226-360, mmu-*miR-29b-3p*, Exiqon) or 1 nM U6 has/mmu/rno miRCURY Detection probe as positive control (#339111, Exiqon) was applied on the slides and incubated for 2 h at 54 °C. Subsequently, the slides were washed six times with SSC-buffer according to the manufacturers schedule and incubated with blocking solution. Next, the anti-DIG reagent (sheep anti-DIG-AP, #11093174910, Roche, Basel, Switzerland) was applied for 1 h at 30 °C, then washed three times for 3 min with PBS. AP substrate (#11697471001 NBT/BCIP ready-to-use-tablets, Roche; #31742, Sigma-Aldrich, St. Louis, MO, USA) was added and incubated for 2 h at 30 °C while protected from light. The following steps were performed exactly as recommended by the supplier using Nuclear Fast Red solution (#N3020, Sigma-Aldrich) for nuclear counter staining. The sections were analyzed by light microscopy (BX61, Olympus, Hamburg, Germany) equipped with a 20× objective (UPlanSApo 20×/0.75, Olympus) the subsequent day after settling overnight.

### 2.4. SDS Gel Electrophoresis and Western Blotting

Proteins were isolated from cervical spinal cord and cerebellar tissue using Cell lysis buffer (#9803S, Cell Signaling Technology, Danvers, USA) supplemented with proteinase inhibitor (#11697498001, Sigma-Aldrich). To determine the obtained protein concentration Pierce^TM^ BCA Protein Assay Kit (#23225, Thermo Fisher Scientific, Waltham, MA, USA) was used. Fifty micrograms of total protein were applied per lane, separated by SDS gel electrophoresis and transferred to nitrocellulose membranes. Blots were blocked in 1 × PBS containing 1% RotiBlock (#A151, Roth) for at least 1 h at RT. For detection of BMF, anti-BMF monoclonal rabbit antibody was used as primary antibody (1:500, #ab181148, Abcam, Cambridge, UK). A secondary horseradish-peroxidase-conjugated goat anti-rabbit antibody (1:5000; #sc-2054, Santa Cruz, Dallas, TX, USA) was used. For the detection of Bax, mouse monoclonal IgG anti-Bax (2D2) (1:200, #sc-20067, Santa Cruz) was used as well as secondary horseradish-peroxidase-conjugated mouse anti-IgGκ light chain immunoglobulin antibody (1:1000, sc-516102, Santa Cruz). Calnexin was used as control protein (1:500; #sc-11397, Santa Cruz). Protein bands were visualized with Western Blotting Luminol Reagent (sc-2048, Santa Cruz). The band intensity was quantified by arithmetic analysis using the software ImageJ. Data were normalized to Calnexin. Normalized values of wobbler samples were compared to normalized wildtype samples at each stage of age and displayed as percentage in a bar chart. Data analyses were performed with GraphPad Prism 6.1 software (GraphPad Software). Data are provided as normalized means ± SEM. Differences were tested using Student’s t-test and results with *p* < 0.01 were considered to be statistically significant.

### 2.5. Immunostaining and Confocal Laser Scanning Microscopy

To investigate the expression of cleaved-Caspase 3 in spinal cord and cerebellum, animals at p40 were anaesthetized with ketamine (100 mg/kg) and xylazine (10 mg/kg) and perfused with 4% PFA in PBS. After decapitation and dissection of the cerebellum and cervical spinal cord, the tissue was post fixated with 4% PFA in PBS at 4 °C for 24 h, transferred to 30% sucrose for 48 h and frozen in isopentane at −45 °C. Tissue was embedded in tissue freezing medium (Cryoglue #30001100, Slee, Mainz, Germany), 16 µm thick slices were cut on a cryostat (Leica AM 3050 S, chamber and stage temperature of −18 °C and −23 °C, respectively) and dried at 37 °C for 30 min.

For immunostaining, slices were incubated with 0.3% Triton-X 100, 5% goat serum in PBS for 60 min. Further the slices were incubated with primary antibodies (1:200; rabbit cleaved-Caspase-3 antibody, #9661, Cell Signaling; rabbit BMF antibody, #ab181148, abcam; mouse Bax antibody, #sc-20067, Santa Cruz) at 4 °C over night. The samples were then reacted with secondary antibodies for 2 h at room temperature (1:200; anti-rabbit-AlexaFluor488, #A11008, Thermo Fisher Scientific; anti-mouse-AlexaFluor488, #A11001, Thermo Fisher Scientific). Nissl staining was performed with NeuroTrace (1:100; #N21482, Thermo Fisher Scientific) for 20 min at room temperature. Nuclei were stained with DAPI (1 µg/mL; #D9542, Sigma-Aldrich). Finally, slices were rinsed in PBS and cover slipped in fluoromount (#F4680, Sigma-Aldrich).

Samples were imaged using an inverted confocal Laser Scanning Microscope (LSM 800, Carl Zeiss Microscopy GmbH, Jena, Germany) equipped with the respective filter sets in combination with a 40× objective (Plan-Apochromat 40×/1.4 Oil, Carl Zeiss Microscopy GmbH). Secondary antibodies were tested for specificity and showed no unspecific binding.

## 3. Results

Since it is known that the cerebellum of wobbler mice shows no signs of neuronal degeneration, this study aimed to determine whether specific mechanisms in the apoptotic pathway of the cerebellum exist that prevent degeneration. Therefore, we investigated different mediators of the intrinsic apoptotic pathway on mRNA and protein level.

### 3.1. Differential Expression of the Proapoptotic Bcl-2-Modifying Factor during Different Stable Symptomatic Stages in Cerebellum and Spinal Cord of Wobbler Mice

*BMF* mRNA expression in the cervical spinal cord and cerebellum of wildtype and wobbler mice was quantified by qPCR at two developmental stages (p0 and p40). In the cerebellum of wobbler mice, mRNA levels of *BMF* were significantly altered at both stages (Figure 1A). While an increased amount of *BMF* mRNA was found in wobbler mice at the presymptomatic stage (p0), mRNA levels were significantly decreased in stable symptomatic stage (p40) (Figure 1A). Thus, it can be concluded that an initially increased expression of *BMF* is found in the cerebellum of the wobbler mice, followed by a decline with age. In comparison, mRNA levels of *BMF* in the spinal cord of wobbler mice displayed no changes at presymptomatic (p0) and stable symptomatic stage (p40, Figure 1A). Protein level of BMF was also found to be differentially expressed in the cerebellum as well as in the spinal cord of wobbler mice (Figure 1B,C). While a significant upregulation in the presymptomatic stage (p0) in the cerebellum could be observed, a significant downregulation of protein expression was identified at the stable symptomatic stage (p40). By contrast, we found a significant downregulation of BMF protein expression in spinal cord of p0 wobbler mice followed by a nonsignificant diminished protein expression at p40. Furthermore, we investigated the localization of the protein BMF in the cerebellum and spinal cord of WT and WR mice by immunostaining. We found a specific staining of isolated BMF-positive cells in the granule as well as in the molecular cell layer of WT cerebella but not in WR animals (Figure 1D). In the spinal cord, a clear staining of BMF can be seen in large cells of the anterior horn in both WT and WR (Figure 1E).

### 3.2. Different Tissue-Specific Modulation of Bax Expression

After it was shown that BMF is differently expressed in certain CNS areas, it should be clarified whether downstream proteins of the intrinsic apoptotic signaling cascade are tissue-specifically modulated. For this purpose, the protein expression of Bax in the stable symptomatic stage (p40) was analyzed. Again, a tissue-specific expression pattern of Bax protein could be determined, with a significant downregulation of Bax in the cerebellum and a slightly elevated but not significant alteration of Bax in the cervical part of the spinal cord of wobbler mice (Figure 2A,B). Here, we also investigated the localization of the protein Bax in the cerebellum and spinal cord of WT and WR mice by immunostaining. We found a specific staining of isolated Bax-positive cells in the granule as well as in the molecular cell layer of WT cerebella but not in WR animals (Figure 2C). In the spinal cord, a clear staining of Bax can be observed in large cells of the anterior horn in both WT and WR (Figure 2D).

### 3.3. Increased Level of Cleaved-Caspase 3 in Motor Neurons of Wobbler Spinal Cord

Furthermore, we investigated the activation status of the effector Caspase 3 in order to find out whether the altered Bax expression also has a downstream effect at the stable symptomatic stage of the disease (p40). Wobbler animals displayed no higher level of cleaved-Caspase 3 in cerebella just like their healthy littermates (Figure 3A,B). The protein levels of the cleaved-Caspase 3 are even slightly lowered (Figure 3A,B). In contrast to this, spinal cords of wobbler mice show more than two-fold elevated levels of cleaved-Caspase 3 compared to wildtype (Figure 3A,B). This result is consistent with the respective pro- or antiapoptotic regulation of members of the Bcl2-family in the intrinsic death cascade. Furthermore, immunohistochemical staining revealed that cleaved-Caspase 3 is clearly localized in large cells with a large nucleus in the ventral horn of the cervical spinal cord of wobbler mice that can be assigned to motor neuronal cells (Figure 3D). Nevertheless, there are scattered cleaved-Caspase 3 positive cells in the granule cell layer of the cerebellum in wildtype mice (Figure 3C).

### 3.4. Differential Expression of miR-29b-3p during Different Stable Symptomatic Stages in Cerebellum and Spinal Cord of Wobbler Mice

Since it is known that noncoding RNAs have a major influence on post-transcriptional regulation, we quantitatively examined the expression of the BMF-targeting miRNA *miR-29b-3p* in the cerebellum and spinal cord of wobbler and control mice. We could show that the *miR-29b-3p* is differentially expressed in cerebella and spinal cord of wildtype and wobbler mice (Figure 4A). Looking at the specific expression in the cerebellum, it becomes apparent that at p0 there is a highly significant downregulation of this miRNA. At p40, however, a highly significant upregulation of *miR-29b-3p* is detectable in cerebella of wobbler mice compared to wildtype (Figure 4A). In the spinal cord, the expression level of *miR-29b-3p* follows a different pattern compared to the cerebellum. Already at birth, a highly significant overexpression of this miRNA is evident in the spinal cord of the wobbler animals (Figure 4A). This overexpression is reversed in the course of the disease, leading to a strongly significant downregulation of *miR-29b-3p* at p40 (Figure 4A).

### 3.5. Expression Pattern of miR-29b-3p in Cerebellum and Spinal Cord

In order to investigate the spatial expression pattern of *miR-29b-3p* in the cerebellum and spinal cord at stable symptomatic stage (p40), we performed in situ hybridization using specific LNA miRNA detection probes. Staining of cerebellar slices revealed that the examined miRNA is mainly expressed in large cells located in the Purkinje cell layer (PCL) in both wildtype and wobbler mice (Figure 4B). Furthermore, expression in small cells located in the white matter (WM) of wobbler mice was detected (Figure 4B). *miR-29b-3p* staining is mainly apparent within the grey matter of the spinal cord of wildtype mice (Figure 4(C1)). It is noticeable that cells with a large cell body, which point to motor neurons, express *miR-29b-3p*. However, especially in the dorsal horn, smaller cells also show positive signals for *miRNA-29b-3p* in wildtype mice (Figure 4(C2)). Wobbler mice show a strongly reduced, hardly recognizable staining of *miR-29b-3p* in grey matter (Figure 4(C3,4)).

## 4. Discussion

ALS is a motor neuron disease, with neurodegeneration and apoptosis taking place in the motor cortex and spinal cord [89,90,91]. The vulnerability of motor neurons in the CNS to pathomechanisms of ALS seems to be very specific. It is arguable and until today the subject of discussions which particular characteristics of motor neurons might cause their fatal susceptibility to degeneration. Alternatively, yet unknown protective mechanisms could exist in other regions of the CNS, the lack of which might expose motor neurons to this particular injury.

The present study was intended to investigate region-specific differences in the intrinsic apoptotic pathway and to provide whether miRNAs might be involved in a degenerative or protective mechanism.

BMF (Bcl2-modyfying factor) is a proapoptotic member of the Bcl2 family. Under physiological conditions, BMF is present in the cytoskeleton of the cytosol bound to myosin V. Pathological noxae cause BMF to detach, translocate, and bind pro- and antisurvival Bcl2-proteins, including Bak and Bax [92,93]. In addition, BMF seems to play a special role in neurodegeneration, since β-amyloid and NGF withdrawal cause its upregulation [78]. BMF acts as a sensitizer and activator at the beginning of the intrinsic apoptotic cascade by binding to Bcl2, thereby inhibiting its prosurvival function [94,95]. Accordingly, BMF positively correlates with apoptosis [94,95]. Several experiments investigated the consequences of BMF knockdown in mice. Its downregulation or knockdown steadily contributed to protection from neuronal cell death, both in vivo and in vitro [96,97,98,99]. Conversely, overexpression was associated with an elevated rate of cell decay [100]. Several studies indicate that coregulation at transcriptional and post-transcriptional level alter BMF expression, triggered by various stimuli [94,97]. *MiR-29b-3p* regulates various members of the Bcl2-family at posttranscriptional level including the antisurvival BH3-only Protein BMF [87].

Using molecular techniques, we were able to show a differential tissue-specific expression of BMF, Bax, cleaved-Caspase3 as well as *miR-29b-3p* in the cerebellum and spinal cord of the wobbler mouse. Already at birth a significantly increased *BMF* mRNA and protein expression in the cerebellum of wobbler mice becomes apparent. In contrast, the expression of *miR-29b-3p* was found to be highly significant downregulated at this time. Thus, the antiapoptotic function of *miR-29b-3p* is heavily suppressed, which correlates with elevated proapoptotic BMF levels. As BMF has a known influence on the intrinsic apoptotic signaling cascade, it is possible that these elevated BMF expression levels may represent a mechanism that has a protective role. It is known that apoptosis in the CNS plays an enormously important physiological role during development [101]. During embryonic and early postnatal development, physiological apoptosis plays a crucial and exceptional role in the cerebellum of all mammals in order to establish the characteristic three-layered structure with their meticulous connectivity [102,103]. Especially, Purkinje cells and granule cells originated from the external granule cell layer with defective migration or aberrant projection seem to be targets of the programmed cell death [104,105]. In case of insufficient apoptosis ectopic, misplaced neurons arise that are detrimental to the whole network by failing the refinement of the neural circuits [101,106,107,108,109,110]. This seems to be confirmed by the fact that various mouse mutants, whose Purkinje cells migrate incorrectly, also display an increased programmed cell death [111,112]. At about day 14, the major period of cerebellar histogenesis involving programmed cell death has come to an end [113]. Opposed to this, programmed cell death of this extent is not to be found in the postnatal spinal cord [114]. Furthermore, it needs to be recognized that the sheer density and absolute quantity of cells located in the cerebellum surpasses any other CNS areas, thus sums up to a more prominent number of decaying cells [115]. According to our observations and the known mechanisms regarding indispensable apoptosis in the development of the cerebellum, it could be speculated that decreased *miR-29b-3p* and the subsequently increased BMF may have a protective function in wobbler cerebella at p0.

Unfortunately, to date, no further research exists that elucidates whether *miR-29b-3p* suppresses BMF protein expression by its mRNA degradation or by blockage of the translation [87]. Thus, two different mechanisms could contribute to the drop in BMF protein expression at p40 in the WR cerebellum: (1) the significantly reduced mRNA level of *BMF* at p40 could be directly caused by the highly significant increase of *miR-29b-3p* expression via degradation by miRNA, finally resulting in a decline of BMF protein; (2) initially lower *BMF* mRNA levels in wobbler cerebella are complemented by an additional blockage of mRNA translation through *miR-29b-3p,* leading to a highly significant decreased expression of BMF at protein level.

In summary, the proapoptotic BMF gene and protein expression of wobbler mice cerebella decrease until they reach a significantly lower expression level than those of healthy mice. This BMF counter-regulation, putatively enhanced by *miR-29b-3p*, might be a first indication why cerebellar neurons suffer less from wobbler mutation induced changes than motor neurons in the spinal cord. Interestingly, by using immunostaining, we were able to clearly demonstrate the localization of the proapoptotic factor BMF in the granule and molecular cell layer of WT cerebella.

In contrast, a reverse development takes place in the spinal cord of wobbler mice. While mutant mice do not show altered *BMF* mRNA at birth, significantly increased *miR-29b-3p* expression could lead to suppression of the BMF protein expression, equating a more neuroprotective situation [87,116]. It could be speculated that these initially protective conditions cannot be maintained since the *miR-29b-3p* level decreases significantly at stable symptomatic stage and thus leads to an indirect upregulation of the BMF protein, albeit not significant. Based on these observations in the spinal cord of wobbler animals, it might be possible that proapoptotic BMF cannot be suppressed by raising the *miR-29b-3p* expression to a favorable level.

To confirm the rather neuroprotective balance in the cerebellum of wobbler mice at p40, we investigated further downstream factors in the apoptotic pathway. BMF itself directly leads to an increase of proapoptotic Bax as well as to a similar suppression of the antiapoptotic Bcl2. Correspondingly, Bax protein levels in wobbler animals are significantly reduced at stable symptomatic stage in the cerebellum, whereas they are found to be slightly elevated in the spinal cord. Furthermore, recent studies have indicated that *miR-29b-3p* also directly reduces Bax expression by degradation of Bax mRNA [93,117,118], reinforcing the prosurvival character of the miRNA. Here, we could also determine a specific localization of Bax in single cells of the granular and molecular cell layer of WT cerebella, which is in line with the previous findings on BMF localization. It is estimated that these are physiological processes in the cerebellum of healthy mice. In the spinal cord, the expression of Bax is found solely in motoneuronal cells of both WT and WR animals, so that the mechanisms investigated can be applied predominantly to these cells.

Consequently, we then assessed cleaved-Caspase 3 protein levels via western blot as an ultimate indicator for the occurrence of apoptosis. Our experiments confirm that this final step of the signaling pathway mirrored previous upstream tendencies either favoring apoptosis or cell survival in wobbler mice. Owing to a consequent suppression of proapoptotic BMF and Bax, the cerebellum of wobbler mice does not suffer from any additional cell decay despite also bearing the detrimental Vps54 mutation. It should be mentioned that a specific localization of the proapoptotic factors, BMF and Bax, applies to single cells of the granule and molecular cell layer of WT animals. Cerebella from wobbler mice do not show any positive signals in immunostainings. Purkinje cells showing a strong expression of *miR-29b-3p* in in situ hybridization experiments do not show BMF or Bax-positive signals in WT or WR animals. Since the *miR-29b-3p* downregulates both BMF and Bax directly, a strong suppression of the protein expression of these two factors would be possible. Thus, the signal intensity of the BMF or Bax fluorescence of Purkinje cells would differ so much from that of BMF or Bax positive cells that they would no longer be detectable. On the contrary, a significant elevation of cleaved-Caspase 3 was detected in the wobbler spinal cord that apparently failed to activate any comparable, protective mechanisms.

In order to clarify which specific cell population primarily perishes, we performed IHC of cleaved-Caspase 3. Hence, particularly spinal cord motor neurons displayed cleaved-Caspase 3 staining, while only scattered positive signals were detectable in the cerebellum of WT, which can be assigned to physiological processes. After we observed that a putative defensive upregulation of the miRNA is absent in spinal cords but present in cerebella, we examined the localization of *miR-29b-3p* via ISH. Against first assumptions, positive probe signaling was found not only in the three neuronal layers of the cerebellum, but also in the white matter of wobbler mice. There solely axons and glia cells, such as astrocytes and microglia, are located [119]. Therefore, *miR-29b-3p* targets not exclusively neuronal mRNAs, but also those of glial cells, emphasizing the interaction of neurodegeneration and neuroinflammation.

*MiR-29b-3p* is counted among the highly conserved *miR-29* family, which additionally embraces *miR-29a* as well as *miR-29c*. The mature microRNA sequence is expressed from two gene clusters that are transcribed from two different chromosomes and finally located in the nucleus [120]. *MiR-29b* is highly expressed in neurons, microglia, astroglia, the brain, cerebellum, and spinal cord [47,121]. Up to now, its impact could not to be fully illuminated. This can be explained in particular by the fact that different mechanisms seem to be involved or related to the regulation of distinct cell populations or pathologies. For instance, *miR-29b* distinguished itself as a critical microRNA in several cancers: in some cases, as an oncogene, while mainly as tumor-suppressor [122,123]. *MiR-29b-3p* is, however, involved in various other processes such as regulation of extracellular matrix, insulin signaling, and angiogenesis [124,125,126]. Increase of *miR-29b-3p* seems to be crucial during neuronal maturation, as it inhibits apoptosis in differentiated neurons [87,127]. Furthermore, alterations in *miR-29b* expression were found in Alzheimer disease and other CNS disorders [128,129]. Intriguingly, brain-specific knockout or deficiency of *miR-29b* caused neuronal cell death and ataxia—symptoms that also apply to wobbler mice—highlighting an outstanding importance for the cerebellum [130,131]. Furthermore, in vivo and in vitro knock down models of *miR-29b* demonstrated loss of neuronal viability in spinal cord injury or dorsal root ganglia [76,126], while delivery of a *miR-29b-3p*-mimic, prohibits chemotherapeutically induced apoptosis in cardiomyocytes through regulation of the mitochondria-dependent apoptotic pathway [117]. The overexpression of *miR-29b-3p* could alleviate or prevent OGD-induced (oxygen-glucose deprivation) damage [132,133], whereas its loss led to aggravation of the situation [134]. As shown in optic nerves of zebra fish, raised *miR-29b-3p* level is believed to play a key role in central nerve regeneration by directly aiming filaments of the cytoskeleton [135]. Despite these observations, the role of *miR-29b* regarding apoptosis, especially in the context of neuroprotection, still remains subject of controversial discussion. In spite of that, there are data indicating a rather proapoptotic nature of *miR-29b-3p*. Namely, *miR-29b-3p* has been described to alter BH3-only protein levels in favor of neuronal degeneration [76,84]. In addition, *miR-29* has an activating effect on the proapoptotic factor p53 [136], which was found to be elevated in both ALS patients and the wobbler mouse [137,138]. However, since in this study we were able to show that the *miR-29b-3p* at p40 is downregulated in the spinal cord in wobbler mice, the increase of p53 does not seem to be due to this mechanism. Moreover, another member of the *miR-29* family, *miR-29a*, whose sequence differs in one nucleotide [120] has been reported to promote Mcl1-mediated cell death [139]. In consequence, a distinct classification among presurgical factors remains highly debatable. Besides *miR-29b-3p*, other miRNAs such as *miR-125b*, *miR-497* or *miR-181a* have also been reported to alter Bcl2-family member’s expression [140,141].

Besides its regulation of Bcl2-proteins, *miR-29b-3p* has been described to exert prosurvival influence through regulation of neuronal iron metabolism, targeting DNA methyl transferase 3a or Aquaporin 4 [77,142,143].

With regard to ALS, *miR-29b-3p* has been shown to be elevated in skeletal muscle of ALS patients [144]. In contrast, in myopathies, in which the muscle tissue primarily degenerates, this miRNA is downregulated, as in Duchenne muscular dystrophy (DMD) [145].

Admittedly, the complex, multilayered regulation within a biological system has to be regarded with great caution. According to database research (targetscan.org, mirtarbase.mbc.nctu.edu.tw), *miR-29b-3p* does not exclusively regulate BMF and Bax, but rather several members of the Bcl2-family. On the other hand, redundant targeting of afore-stated ones happens by various microRNAS (*miR-24*, *miR-124*, *miR-128*) and could potentially compensate or antagonize adjustments of *miR-29b-3p*. Therefore, striking alterations might only be caused by a collective interaction of all players involved.

## 5. Conclusions

In this study, we were able to find evidence to answer the question why motor neurons selectively degenerate in amyotrophic lateral sclerosis, while other cells and systems, such as the cerebellum, are not affected. As shown in the overview of our findings in Figure 5, expression studies revealed that several members of the intrinsic apoptotic signaling cascade are expressed differently in the cerebellum and spinal cord of wobbler mice compared to wildtype mice. Additionally, we showed that the different regulations of the apoptotic mediators in the respective tissue correlate with different expression levels of *miR-29b-3p*. In the cerebellum of wobbler mice, *miR-29b-3p* is upregulated at p40 compared to wildtype mice. In consequence, this overexpression downregulates the proapoptotic factors BMF, Bax, and Bak, which leads to neuroprotection. In the spinal cord, however, this miRNA is downregulated so that there is not sufficient inhibition of the translation of proapoptotic factors, leading to apoptosis and thus neurodegeneration. Based on our findings, it could be speculated that *miR-29b-3p* might regulate antiapoptotic survival mechanisms in CNS areas that are not affected by neurodegeneration in the wobbler mouse ALS model.

## Figures and Tables

**Figure 1 cells-08-01077-f001:**
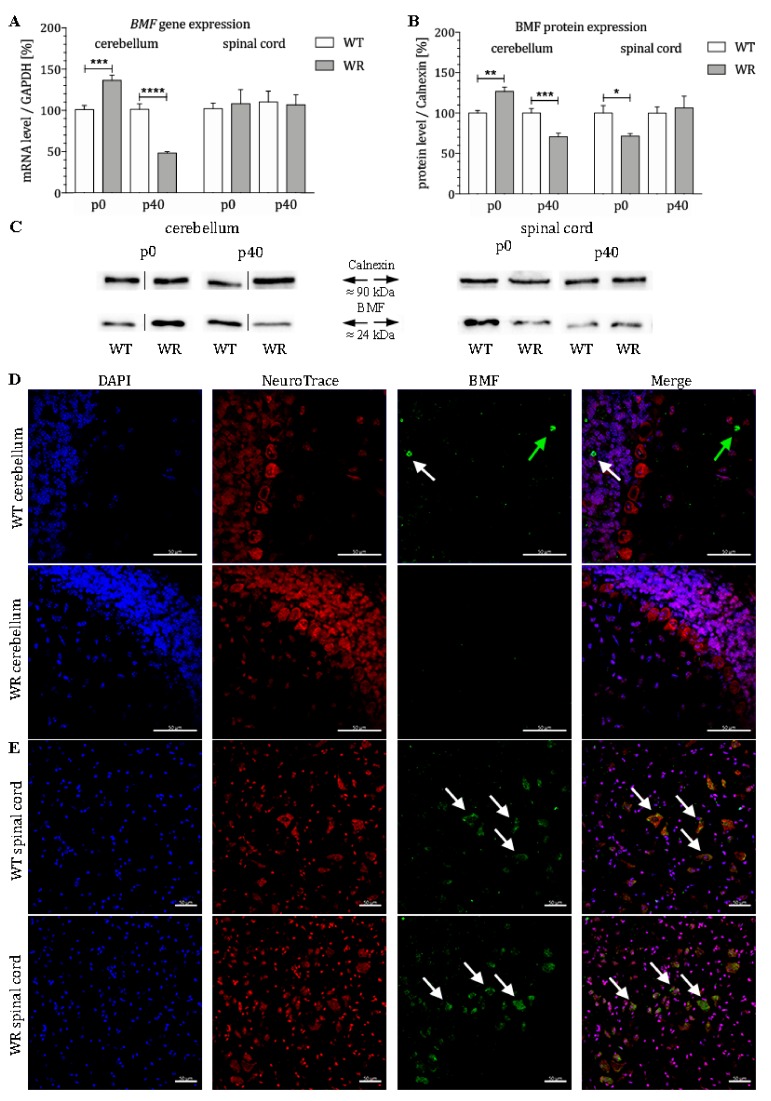
Age-dependent differential expression of BMF in cerebellum and spinal cord of wildtype and wobbler mice. (**A**). *BMF* mRNA levels at stages p0 and p40 in cerebellum and spinal cord of WT and WR. mRNA expression of *BMF* from presymptomatic (p0) to stable symptomatic phase (p40) in cerebellum and spinal cord. qPCR was performed using five samples for each genotype and stage. In cerebellum, a significant upregulation of *BMF* could be detected at stage p0, while a highly significant downregulation could be shown at stage p40. In contrast, no significant regulation could be detected at stages p0 and p40 in spinal cord. For relative quantification of *BMF* expression, the 2^−∆∆Ct^ method was conducted using the housekeeping gene *GAPDH* for normalization; data are provided as means ± SEM. Data were tested for significance using Student’s t-test. Significant differences are indicated by * *p* < 0.05, ** *p* < 0.01, *** *p* < 0.001, **** *p* < 0.0001; *n* = 5. (**B**). Quantitative analysis of BMF protein expression level in the cerebellum and cervical spinal cord from presymptomatic and stable symptomatic phase of WT and WR mice. At stage p0, a significant upregulation in cerebellar tissue was detected, whereas a highly significant decrease of the BMF protein level appears at stable symptomatic stage p40. In the spinal cord, a significant downregulation at p0 could be observed, while no significantly altered BMF expression could be shown at p40. Data are provided as means ± SEM. Data were tested for significance using Student’s t-test. Significant differences are indicated by * *p* < 0.05, ** *p* < 0.01, *** *p* < 0.001; *n* = 5. (**C**). Exemplary Western blot of BMF protein expression levels in WT and WR cerebellum and spinal cord at developmental stages p0 and p40. Calnexin was used as control protein. Intensity values of BMF were normalized to calnexin and compared to WT tissue. (**D**). Exemplary immunofluorescence staining of NeuroTrace, as Nissl marker (red), and BMF (green). Scattered staining of BMF could be detected within the granular (white arrow) and molecular (green arrow) layer of the cerebellar slices of WT mice at p40. (**E**). Exemplary immunofluorescence staining of NeuroTrace, as Nissl marker (red), and BMF (green). Specific staining for BMF was found to be located in NeuroTrace positive cells with a large cell body and nucleus, that can be assigned to be motor neuronal cells (white arrows). Nuclei were stained with DAPI (blue). Pictures were taken using a confocal laser scanning microscope (LSM 800, Zeiss, Germany with a 40× objective (PlanApo 40×/1.4 Oil DICII, Nikon instruments). Scale bar = 50 µm.

**Figure 2 cells-08-01077-f002:**
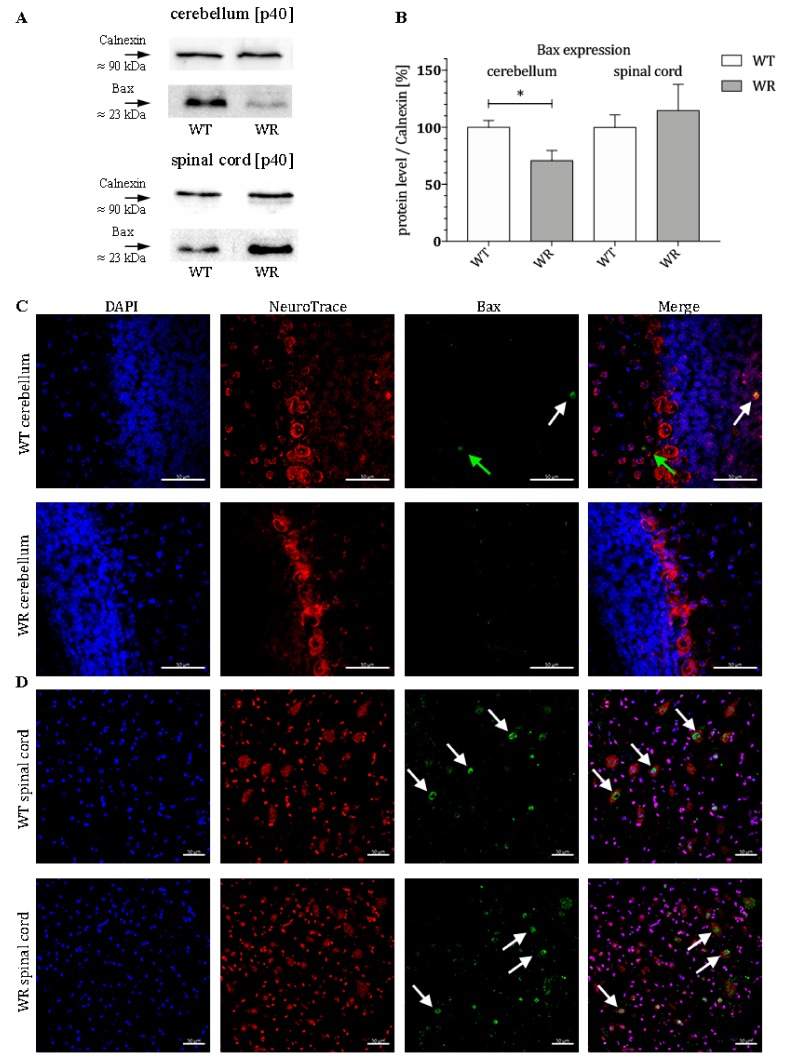
Differential expression of Bax in cerebellum and spinal cord of wildtype and wobbler mice. (**A**). Exemplary Western blot of Bax protein expression levels in WT and WR cerebellum and spinal cord at stable symptomatic stage p40. Calnexin was used as control protein. Intensity values of Bax were normalized to calnexin and compared to WT tissue. (**B**). Quantitative analysis of Bax protein expression level in the cerebellum and cervical spinal cord from stable symptomatic phase of WT and WR mice. A significant downregulation was found in developmental stages p40 in the cerebellum, while there is no significant deregulation in the spinal cord. Data are provided as means ± SEM. Data were tested for significance using Student’s t-test. Significant differences are indicated by * *p* < 0.05. *n* = 6. (**C**). Exemplary immunofluorescence staining of NeuroTrace, as Nissl marker (red), and Bax (green). Scattered staining of Bax could be detected within the granular (white arrow) and molecular (green arrow) layer of the cerebellar slices of WT mice at p40. (**D**). Exemplary immunofluorescence staining of NeuroTrace, as Nissl marker (red), and Bax (green). Specific staining for Bax was found to be located in NeuroTrace positive cells with a large cell body and nucleus, that can be assigned to be motor neuronal cells (white arrows). Nuclei were stained with DAPI (blue). Pictures were taken using a confocal laser scanning microscope (LSM 800, Zeiss, Germany with a 40× objective (PlanApo 40×/1.4 Oil DICII, Nikon instruments). Scale bar = 50 µm.

**Figure 3 cells-08-01077-f003:**
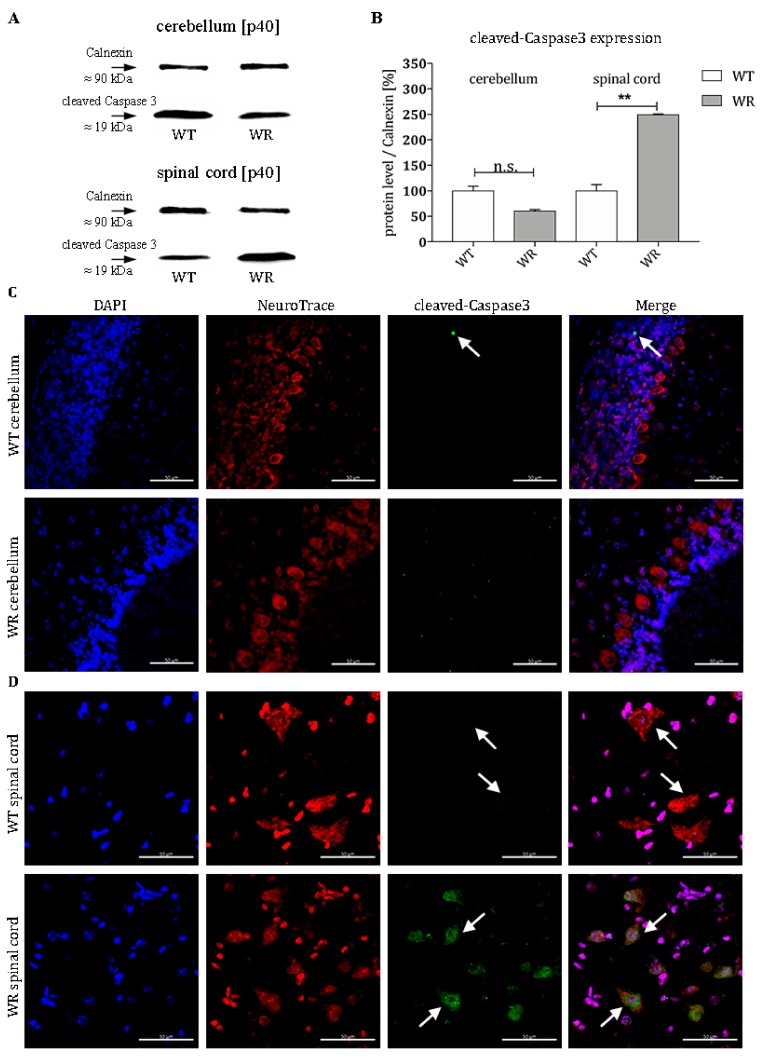
Alterations in cleaved-Caspase 3 expression in cerebellum and spinal cord of wildtype and wobbler mice. (**A**). Exemplary Western blot of cleaved-Caspase 3 in WT and WR cerebellum and cervical spinal cord at developmental stage p40. Calnexin was used as control protein. Intensity values of cleaved-Caspase 3 were normalized to calnexin and compared to WT tissue. (**B**). Quantitative analysis of cleaved-Caspase 3 protein expression level in the cerebellum and cervical spinal cord from stable symptomatic phase of WT and WR mice. No significant regulation was found in the cerebellum in wobbler mice, while there is a significant upregulation in the spinal cord. Data are provided as means ± SEM. Data were tested for significance using Student’s t-test. Significant differences are indicated by n.s. = not significant, ** *p* < 0.01; *n* = 6. (**C**). Exemplary immunofluorescence staining of NeuroTrace, as Nissl marker (red), and cleaved-Caspase 3 (green). Nuclei were stained with DAPI (blue). Scattered staining of cleaved-Caspase 3 could be detected within the granular layer of the cerebellar slices of WT mice at p40 (white arrow). (**D**). Exemplary immunofluorescence staining of NeuroTrace, as Nissl marker (red), and cleaved-Caspase 3 (green). Nuclei were stained with DAPI (blue). Specific staining for cleaved-Caspase 3 was found to be located in NeuroTrace positive cells with a large cell body and nucleus, that can be assigned to be motor neuronal cells (white arrows). In WT cells the staining of cleaved-Caspase 3 was less marginal, than in WR cells. Pictures were taken using a confocal laser scanning microscope (LSM 800, Zeiss, Germany with a 40× objective (PlanApo 40×/1.4 Oil DICII, Nikon instruments). Scale bar = 50 µm.

**Figure 4 cells-08-01077-f004:**
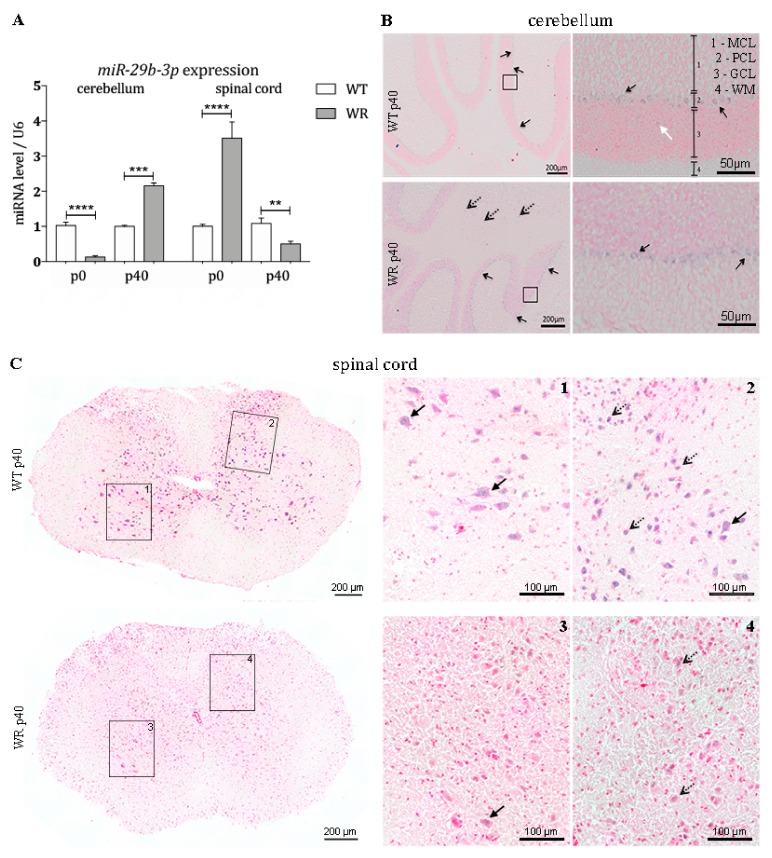
Age-dependent differential expression of *miR-29b-3p* in cerebellum and spinal cord of wildtype and wobbler mice. (**A**). Expression levels of *miR-29b-3p* from presymptomatic to stable symptomatic phase of wildtype (WT) and wobbler (WR) cerebella and spinal cords were investigated by qPCR. Significant deregulation of *miR-29b-3p* was detected at all developmental stages. For relative quantification of *miR-29b-3p* expression, the 2^−ΔΔCt^ method was conducted using the housekeeping gene U6 for normalization. Data are provided as means ± SEM. Data were tested for significance using Student’s t-test. Significant differences are indicated by ** *p* < 0.01, *** *p* < 0.001, **** *p* < 0.0001; *n* = 6. (**B**). Exemplary overview of in situ hybridization with *miR-29b-3p*-probe in a sagittal section of the cerebellum of wildtype (WT) and wobbler (WR) mice at the developmental stage p40. Distinct signals can be found in large cells within the Purkinje cell layer (PCL), indicating Purkinje cells (arrows with closed head). Scattered signals can be found in the granular cell layer (white arrow). In sections of wobbler mice, additional signals can be found in small cells within the white matter (WM) of the cerebellum (arrows with open head). Size of scale bars are indicated in pictures. (**C**). Exemplary overview of in situ hybridization with *miR-29b-3p*-probe in a cross section of the cervical spinal cord of wildtype (WT) and wobbler (WR) mice at the developmental stage p40. *miR-29b-3p* staining is mainly apparent within the grey matter of the spinal cord of wildtype mice. Cells with a large cell body, which point to motor neurons (arrows with closed head), express *miR-29b-3p*. Additional signals can be found in small cells especially in the dorsal horn of the spinal cord (arrows with open head). Wobbler mice show a strongly reduced staining of *miR-29b-3p* in grey matter. Size of scale bars are indicated in pictures. All pictures were taken with a light microscope (Olympus microscope BX61VS, Japan) and a 20× objective (UPlanSApo 20×/0.4, Olympus, Japan). For counterstaining, Nuclear Fast Red was used.

**Figure 5 cells-08-01077-f005:**
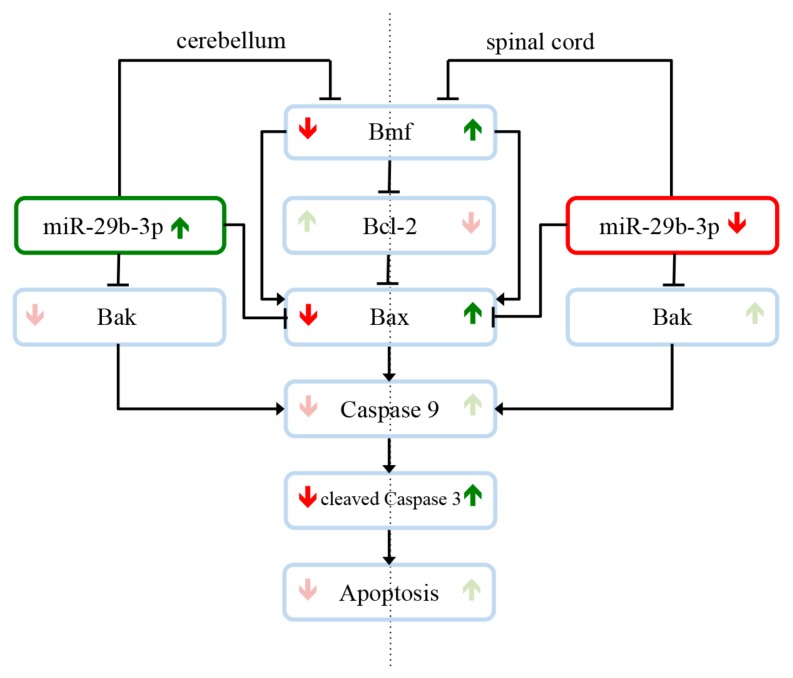
*miR-29b-3p*-mediated neuroprotection and neurodegeneration in the Wobbler mouse in the stable symptomatic phase (p40). Expression studies revealed that several members of the intrinsic apoptotic signaling cascade are expressed differently in the cerebellum and spinal cord of wobbler mice compared to wildtype mice. Different regulations of the apoptotic mediators in the respective tissue correlate with different expression levels of *miR-29b-3p*. In the cerebellum of wobbler mice, *miR-29b-3p* is upregulated at p40 compared to wildtype mice. This overexpression downregulates in the following the proapoptotic factors BMF, Bax, and Bak, which leads to less apoptosis and thus neuroprotection. In the spinal cord, however, *miR-29b-3p* is downregulated so that there is not sufficient inhibition of the translation of proapoptotic factors, leading to apoptosis and thus neurodegeneration. These results suggest that *miR-29b-3p* regulates antiapoptotic survival mechanisms in CNS areas that are not affected by neurodegeneration.

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
