# Peer review of "Deregulated miR-29b-3p Correlates with Tissue-Specific Activation of Intrinsic Apoptosis in An Animal Model of Amyotrophic Lateral Sclerosis"

_cells, 2019, doi:10.3390/cells8091077_

Round 1
Reviewer 1 Report
The topic addressed is one of the leading themes in neurodegenerative diseases, the role of microRNAs (miRNAs). The data provided are interesting and add to the current knowledge, especially addressing the issue of the differential penetration of the neurodegenerative processes in distinctive brain compartments in ALS. The authors have recently published on another miRNA misregulation in the cervical part of the wobbler spinal cord (Rohm M, May C, Marcus K, Steinbach S, Theis V, Theiss C, Matschke V. The microRNA miR-375-3p and the Tumor Suppressor NDRG2 are Involved in Sporadic Amyotrophic Lateral Sclerosis. Cell Physiol Biochem. 2019;52(6):1412-1426). In that manuscript they look at all the stages linked to disease progression p0, p20, p40 showing differences in all of them. Hence, although they offer a rationale about why in the present manuscript they consider only two points (p0 and p40) I think it would be interesting to analyze all the time points. How sure are the authors the animals did not show subtle changes already at two months? Moreover, since both studies are linked to the induction/suppression, control of the apoptotic processes they should ponder these aspects in their discussion.
Further aspects:
-why they use just a marker for glia? I tend to agree that motor neurons are easily recognizable, because of their cell body size, yet they should probe the motor neurons with specific markers to positively recognize them.
-the authors use calnexin as standard in their WB experiments, I’m not familiar with this standard I know though that calnexin is a chaperon protein involved in some ER pathways. Is a good standard?
-I wouldn’t use clinical, as in “clinical stages”.
Reviewer 2 Report
Klatt and colleagues describe in the cerebellum and cervical spinal cord of the wobbler mouse, a sporadic ALS model, a differential expression of members of the intrinsic apoptotic pathway and the miRNA miR-29b-3p. The data shown is in line with published work by other groups on the roles of apoptosis and miRNAs in the disorder. The originality of the manuscript is i) the analysis and comparison of the cerebellum and the spinal cord and ii) the differential expression of the specific miR-29b-3p with members of the intrinsic apoptotic pathway in the two tissues. Nevertheless, a clear mechanism behind the role of miR-29b-3p in regulating apoptosis is not explored.
The manuscript could be strengthened by:
Modification of the title: the current title is misleading. The work submitted is only descriptive and lacks any mechanism to support that neuroprotection and neurodegeneration are mediated by miR-29b-3p. The introduction could be revised to better describe the state of the art on the connection between ALS, apoptosis and miRNA. Additional references on some of the most recent articles on ALS and miRNA could be added (for instance Rizzuti et al. 2018). Moreover, the reader could be helped by a better description of the wobbler mouse (with its pros and cons). References should be revised. References 19, 20, 21, 22 and 33 are likely not correct. Figure 1: western blots (WBs) in C seem to be all from different blots (P0 and P40 samples are likely not on the same blots), if correct, please highlight this fact with borders on every single blot. Figure 2: the labelling of the WBs could be improved (WT and WR, although present in figure 1, are missing). Moreover, it would be good to also show the levels of the proteins at p0. Figure 3: WBs need better labelling, as in figure 2. C and D need neuronal markers as usually done in the majority of the other publications in the field. Figure 4: B, analysis of p0 could strengthen the findings. Although not easy, a cellular marker should be added to the analysis. Please check Nielsen et al. 2011. In addition, manipulation of expression of the miRNA miR-29b-3p with mimics or inhibitors would improve the manuscript immensely (see Nolan et al. 2014 for reference). In the discussion, please revise the text to highlight the lack of mechanism. There is no direct proof that miR-29b-3p directly regulates anti-apoptotic survival mechanisms in CNS areas that are not affected by neurodegeneration in the ALS model, wobbler mouse.Author Response
Please see the attachment.

Reviewer 3 Report
The paper by Klatt et al. aims to investigate the molecular mechanisms that render motor neurons specifically susceptible to neurodegeneration in Amyotrophic Lateral Sclerosis. The major focus is on the anti-/pro-apoptotic pathways that could modulate neuronal survival. To do so, the authors take advantage of the wobbler mouse model of motor neuron degeneration. The authors analyze the mRNA and protein expression of few members of the apoptotic pathway and of their upstream miRNA (miR-29b-3p) in the spinal cord (a CNS region susceptible to neurodegeneration) versus the cerebellum (a region relatively spared by the disease process), highlighting interesting differences that could account for higher susceptibility of spinal cord motor neurons to the neurodegeneration.
Overall the topic is of interest for the field and the findings could open new opportunities for development of ALS therapeutics. However, there are several concerns on the way the research is presented and the results are described.
Major concerns:
1. The manuscript requires a thorough proofreading to check for typos and english spelling.
Examples:
line 38; "Riluzol" should be "Riluzole"
line 38; "Edavaron" should be "Edaravone"
line 193; "Therefor" should be "Therefore"
line 360; "of this extend" should be "of this extent"
line 400; "abscent" should be "absent"
line 368 is not clear: "caused by the highly significant increase of miR-29b-3p expression via degradation of miRNA..". Please rephrase or clarify.
2. Title. There is large debate on whether wobbler mouse could be considered as a mouse model for sporadic ALS. In fact, the neurodegeneration in this mouse model is due to a spontaneous mutation that occurred in the gene Vps54. Unfortunately, there is lack of association between mutations in VPS54 and ALS in humans. However, the molecular mechanisms leading to motor neuronal loss in wobbler mouse recapitulate to some extent some of the defects that have been found in ALS patients and other mouse models of familiar ALS. Thus, I suggest not to use the word "sporadic" in the definition of this mouse model; simply describing wobbler mouse as "a mouse model of ALS" would be more appropriate.
3. Introduction. It is very difficult to understand the rational of this study by reading the introduction in the way it has been presented. The authors should rephrase the paragraphs to highlight better: i) the features of the wobbler mouse model, including details on the gene mutation causing the disease; ii) reason for choosing the cerebellum as CNS area to be compared with the spinal cord; iii) better description of the downstream targets of miR-29b-3p (highlighting the ones that are pro-apoptotic and the ones that are anti-apoptotic).
4. Results. It is difficult to follow the order of the experiments presented. The manuscript would greatly benefit from a better organization of the order of the results.
e.g. in the title and the introduction the authors focus the attention on miR-29b-3p but then the first results presented are BMF and Bax expression (both pro-apoptotic). No analysis of anti-apoptotic molecules downstream of miR-29b-3p is presented. Analysis of miR-29b-3p expression and its distribution is very interesting but it is presented only at the end of the manuscript.
Moreover, while GAPDH is used as housekeeping gene to normalize mRNA expression levels, calnexin is used as normalizer for western blot analyses. The authors should justify the choice of calnexin, as opposed to actin, tubulin or GAPDH, which are commonly used as normalizers.
5. Discussion. At line 350 the authors speculate that "protective increased apoptosis may occur in wobbler cerebella". This sentence is not clear and the results presented in this manuscript do not support that. Please rephrase or clarify.
Other concerns:
Figure 1.
Fig.1 A shows the expression levels of BMF (mRNA) as ratio to wild type levels whereas Fig. 1B shows the protein levels of BMF as percentage of wild type levels. Please reconcile by choosing a uniform modality of presentation.
Are there any age-related changes in the mRNA and protein levels of BMF? Showing the absolute levels of BMF in the wild type mice at p0 and p40 could help answering this question.
Figure 2.
The levels of Bax are shown only at p40. It would be useful to show the levels of Bax at p0 (like was done for BMF), to see whether changes in Bax parallel the changes previously observed for BMF.
Figure 3.
Fig.3A. Same as Fig. 2. The levels of cleaved caspase 3 are shown only at p40. It would be useful to show the levels at p0, too.
Additionally, Fig. 3A shows detectable levels of cleaved caspase 3 in the wild type, however no positive signal for cleaved caspase 3 is detected by immunofluorescence in Fig. 3C. Any guess on the reasons for this discrepancy?
Fig. 3D. The authors highlight positive signal for cleaved caspase 3 in cells with large nuclei in the spinal cord of wobbler mice. The authors claim that these cells could be motor neurons based on the dimension of their nuclei. Co-localization with GFAP (astrocyte marker) is not appropriate to support this claim. The authors should show the co-localization with Choline-Acetyl transferase marker (ChAT) or with Neurotrace (fluorescence NISSL stain).
Moreover, Fig. 4 highlights a differential regulation and distribution of miR-29b-3p in cerebellum and spinal cord of wobbler mice. It would be critical to show the anatomical and cellular distribution of BMF and Bax, as well. Immunofluorescence staining for BMF and Bax is strongly suggested, to support the claims of this mamnuscript.
Round 2
Reviewer 1 Report
The authors have fully reply to all my comments
Reviewer 2 Report
The authors have addressed most of my comments and concerns.
Reviewer 3 Report
I read the paper and I found it significantly improved. I don't have major concerns for the publication of this manuscript.